# Presence of myxoid stromal change and fibrotic focus in pathological examination are prognostic factors of triple-negative breast cancer: Results from a retrospective single-center study

Hirotsugu Yanai[1], Katsuhiro Yoshikawa[1,2], Mitsuaki Ishida[2]*, Koji Tsuta[2], Mitsugu Sekimoto[1], Tomoharu Sugie[1]

1 Department of Surgery, Kansai Medical University, Osaka, Japan, 2 Department of Pathology and Clinical Laboratory, Kansai Medical University, Osaka, Japan

* ishidamt@hirakata.kmu.ac.jp

**Data Availability Statement:** All relevant data are in the paper and the Supporting Information files.

## Abstract

### Background

Stromal reaction is an important prognostic factor in several cancers, and the presence of myxoid change was assessed as a poor prognostic factor in colorectal cancer. However, the prognostic significance of myxoid change in triple-negative breast cancer (TNBC) remains unknown. This study aimed to determine the prognostic significance of myxoid change and fibrotic focus (FF), which is a fibrotic area within the tumor and considered a poor prognostic indicator in patients with TNBC.

### Methods

We enrolled 62 patients with TNBC and reviewed the surgically resected specimens to evaluate myxoid change and FF in the tumor using previously outlined criteria. We evaluated tumor-infiltrating lymphocytes (TILs) using hematoxylin and eosin slides. Overall survival (OS) and relapse-free survival (RFS) were compared based on the presence of myxoid change and/or FF, and the risk factors for RFS were analyzed.

### Results

Myxoid change and FF were observed in 25.8% and 33.9% of specimens, respectively. Based on stromal lymphocyte infiltration, 19 patients (30.6%) had high TILs, while the remaining 43 patients (69.4%) had low/intermediate TILs. Presence of myxoid change was significantly correlated with poor OS and RFS ($p = 0.040$ and $0.031$, respectively). FF was also significantly correlated with poor OS and RFS ($p = 0.012$ and $0.028$, respectively). The combination of myxoid change and FF was an independent and poor prognostic factor according to the multivariate analysis (HR 11.61; 95% CI 1.027–131.2; $p = 0.048$). Presence

**Funding:** The authors received no specific funding for this work.

**Competing interests:** The authors have declared that no competing interests exist.

of myxoid change and FF were significantly associated with low/intermediate TILs in the stroma ($p = 0.013$).

## Conclusions

Histopathological assessment of myxoid change and FF in TNBC may be a useful, practical, and easily assessable method for predicting prognosis in patients with TNBC, which should be confirmed in larger prospective studies. Diagnostic criteria for the establishment of myxoid change and FF in TNBC must be established, and their underlying molecular events must be clarified.

## Introduction

Recent studies have highlighted the important role of the tumor microenvironment in growth, invasion, and metastasis of cancer [1]. The tumor microenvironment consists of a variety of cells, including inflammatory cells, fibroblasts, myofibroblasts, and endothelial cells, and the extracellular matrix [1]. It has been recognized that cancer-associated fibroblasts are one of the main components of the stromal reaction of carcinoma, and they play an important role in constituting the tumor microenvironment through the secretion of cytokines and extracellular matrix [1, 2]. Moreover, cancer-associated fibroblasts have been reported to exhibit prognostic implications in various types of carcinomas, including breast cancer [2–4].

More or less, stromal reaction is frequently observed in the invasive carcinoma tissues [5]. Recent studies have revealed that the patterns of stromal reaction (desmoplastic reaction) due to carcinoma invasion play an important role in the prognosis of cancer patients [5]. The prognostic significance of the desmoplastic reaction has been characterized in colorectal cancer [6, 7]. The patterns of desmoplastic reaction were classified as mature, intermediate, or immature, depending on the presence of myxoid stroma and hyalinized collagen in the invasive front of the tumor. These patterns were identified as independent prognostic factors for recurrence-free survival in patients with colorectal cancer. While mature group had the highest recurrence-free survival rates, the immature group (presence of myxoid change) had the worst prognosis [6, 7]. Furthermore, stromal reaction routinely occurs in breast cancer tissues. However, the prognostic significance of the different stromal reaction patterns in breast cancer tissues remains inadequately evaluated [4, 8]; nevertheless, it has been reported that the presence of fibrotic focus (FF), characterized by a fibrotic area containing fibroblasts and collagen, which is surrounded by a cellular zone of infiltrating carcinoma, is a poor prognostic factor in patients with breast cancer [9, 10]. Triple-negative breast cancer (TNBC) is characterized by the lack of estrogen and progesterone receptors and human epidermal growth factor receptor 2 (HER2). It is known to be one of the most aggressive subtypes of breast cancer. However, the significance of stromal reaction patterns, particularly in TNBC, remains unknown. Thus, the aim of this study was to determine the prognostic significance of the presence of myxoid change and FF in patients with TNBC.

## Materials and methods

### Patient selection

We selected 165 consecutive patients with TNBC who underwent surgical resection at the Department of Surgery of Kansai Medical University Hospital between January 2006 and

December 2018. Patients who received neoadjuvant chemotherapy or those who had a special type of invasive carcinoma were excluded; 62 patients with TNBC were included.

This study was conducted in accordance with the principles of the Declaration of Helsinki and was approved by the Institutional Review Board of the Kansai Medical University Hospital (Approval #2019234). Informed consent was obtained from patients through the opt-out methodology, owing to the retrospective design of the study, with no new risk to the participants. Information regarding this study, such as the inclusion criteria, and the opportunity to opt out was provided through the hospital's website.

### Histopathological analysis

Surgically resected specimens were fixed with formalin, sectioned, and stained with hematoxylin and eosin. More than 2 experienced diagnostic pathologists independently evaluated all histopathological diagnoses. Myxoid change was histopathologically characterized by the presence of amorphous extracellular substances, which comprise an amphophilic or slightly basophilic material within the fibrous stroma, similar to the criteria reported for colorectal cancer [6, 7]. The presence of myxoid change was evaluated in the invasive front and within the tumor. Moreover, the presence of FF in breast cancer was also evaluated according to the previously reported criteria, which was diagnosed when there was a fibrotic area within the tumor containing fibroblasts and collagen, surrounded by a highly cellular zone of infiltrating carcinoma and measuring more than 1 mm [10]. In addition, we also evaluated the presence of myxoid change and FF using the matched pre-operative biopsy specimens.

Furthermore, we evaluated tumor-infiltrating lymphocytes (TILs) using hematoxylin and eosin slides according to the recommendation of the TIL Working Group [11]. We classified ≥60% stromal TILs as high TILs, and <60% stromal TILs as low/intermediate TILs [12]. Assessments of myxoid change, FF, and TILs were independently performed by more than 2 experienced researchers, after reviewing all surgically resected slides.

We used the American Joint Committee on Cancer (8th edition)/Union for International Cancer Control (9th edition) TNM classification and stage groupings. Histopathological grading was based on the Nottingham histological grade [13]. According to a meta-analysis on patients with TNBC, a Ki-67 labeling index (LI) of 40% or more was considered high [14].

### Statistical analyses

All analyses were performed using SPSS Statistics 25.0 (IBM, Armonk, NY, USA). The association between groups was evaluated using Fisher's exact test for categorical variables and Mann–Whitney U test was used for continuous variables. The overall survival (OS) and relapse-free survival (RFS) rates were evaluated using a Kaplan–Meier analysis; log-rank tests were used to compare between groups. The Cox proportional-hazards model was used to examine the relationship between clinicopathological parameters and survival. Agreement between two groups was analyzed using the Kappa test. $P < 0.05$ was considered to indicate statistical significance.

## Results

### Patient characteristics

This study comprised 62 female patients. The patients' clinical characteristics are summarized in Table 1. The median age at initial diagnosis was 68 years (range: 31–93 years). All patients were diagnosed with TNBC according to histopathological findings, and all samples were diagnosed as "invasive carcinoma, no special type". The median tumor diameter was 20 mm

**Table 1. Clinical characteristics of patients with triple-negative breast cancer.**

| Factors | N | % |
|---|---|---|
| Total | 62 | |
| Age (years) | | |
| Median (range) | 68 (31–93) | |
| Menopausal status | | |
| Premenopausal | 9 | 14.5 |
| Postmenopausal | 52 | 83.9 |
| Unknown | 1 | 1.6 |
| Body mass index | | |
| Median (range) | 23.2 (16.2–32.2) | |
| Tumor size (mm) | | |
| Median (range) | 20 (2–55) | |
| Pathological stage | | |
| I | 26 | 41.9 |
| IIA | 23 | 37.1 |
| IIB | 5 | 8.1 |
| IIIA | 4 | 6.5 |
| IIIB | 3 | 4.8 |
| IIIC | 1 | 1.6 |
| Lymph node status | | |
| Positive | 14 | 22.6 |
| Negative | 33 | 53.2 |
| Not tested | 15 | 24.2 |
| Lymphatic invasion | | |
| Positive | 53 | 85.5 |
| Negative | 9 | 14.5 |
| Venous invasion | | |
| Positive | 37 | 59.7 |
| Negative | 25 | 40.3 |
| Nottingham histological grade | | |
| 1 | 2 | 3.2 |
| 2 | 28 | 45.2 |
| 3 | 32 | 51.6 |
| Ki-67 labeling index (LI) | | |
| High | 37 | 59.7 |
| Low | 21 | 33.9 |
| Not tested | 4 | 6.5 |
| Stromal tumor-infiltrating lymphocytes | | |
| High | 19 | 30.6 |
| Low/intermediate | 43 | 69.4 |
| Adjuvant chemotherapy | | |
| Performed | 34 | 54.8 |
| Not performed | 25 | 40.3 |
| Undetermined | 3 | 4.8 |

(range: 2–55 mm). Patients were staged as follows: I (26 patients), IIA (23 patients), IIB (5 patients), IIIA (4 patients), IIIB (3 patients), and IIIC (1 patient). Based on the histopathology, 2, 28, and 32 patients were graded 1, 2, and 3, respectively. The Ki-67 LI was high, low, and

not-tested among 37, 21, and 4 patients, respectively. Based on stromal lymphocyte infiltration, 19 patients (30.6%) were classified as high TILs, while the remaining 43 patients (69.4%) were classified as low/intermediate TILs. The observation period was up to 60 months in all the patients, during which 9 (14.5%) patients experienced relapse (all had distant metastasis, and none experienced local recurrence), 7 (11.3%) patients died due to the disease, and 3 (4.8%) patients died of unrelated causes.

## Association between myxoid change and survival

Myxoid change was observed in 16 patients (25.8%) in this cohort (Fig 1A and 1B). Fig 1C shows that there was only fibrotic stroma without myxoid change around the tumour nests. Table 2 summarizes the clinicopathological factors in myxoid change-positive and–negative

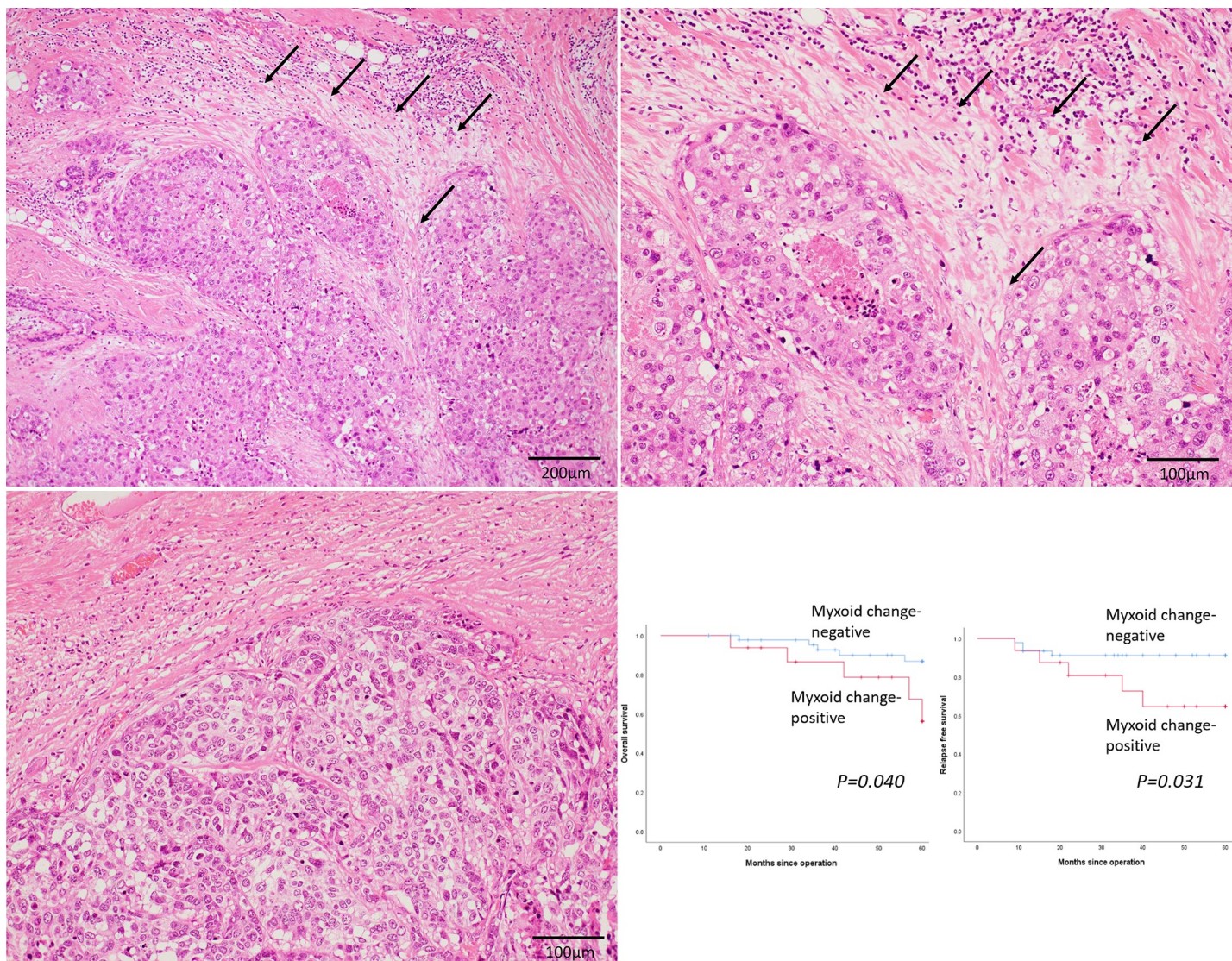

**Fig 1. Myxoid change as an independent predictor of survival in patients with triple-negative breast cancer.** (a) (b) Representative image of myxoid change in triple-negative breast cancer. Accumulation of amorphous stromal substances around the tumor nests (arrows) (x 100 (a) and x 200 (b)). (c) Only fibrous stroma is present, and no myxoid change is observed around the tumor nests (x 200). (d) Kaplan–Meier curves for the overall survival (OS) and relapse-free survival (RFS) of patients with triple-negative breast cancer. OS (left) and RFS (right) curves among myxoid change-positive (red line) and myxoid change-negative (blue line) patients.

patients. Myxoid change was significantly associated with low/intermediate TILs in the stroma (*p = 0.001*). However, there were no significant differences in pathological stage, lymph node status, and Nottingham histological grades between the groups. Fig 1D shows the OS and RFS curves of myxoid change-positive and -negative patients. Presence of myxoid change was significantly correlated with both poor OS (*p = 0.040*) and RFS (*p = 0.031*) in patients with TNBC.

## Association between FF and survival

FF was noted in 21 patients (33.9%) in this cohort (Fig 2A and 2B). Fig 2C shows tumor nests without FF. Table 3 summarizes the clinicopathological factors in FF-positive and -negative

**Table 2. Association of myxoid change with clinicopathological factors.**

| Factors | Myxoid change -positive (n = 16) | Myxoid change -negative (n = 46) | *P*-value |
|---|---|---|---|
| Age (years; median ± standard deviation) | 67±17 | 68±14 | 0.910 |
| Menopausal status | | | |
| Premenopausal | 3 | 6 | 0.686 |
| Postmenopausal | 13 | 39 | |
| Unknown | 0 | 1 | |
| Body mass index | | | |
| | 21.1±3.3 | 23.9±3.6 | *0.017* |
| Tumor size (mm) | | | |
| ≦20 | 5 | 27 | 0.083 |
| >20 | 11 | 19 | |
| Pathological stage | | | |
| I + II | 12 | 42 | 0.187 |
| III | 4 | 4 | |
| Lymph node status | | | |
| Positive | 4 | 10 | 0.456 |
| Negative | 6 | 27 | |
| Not tested | 6 | 9 | |
| Lymphatic invasion | | | |
| Positive | 16 | 37 | 0.096 |
| Negative | 0 | 9 | |
| Venous invasion | | | |
| Positive | 13 | 24 | 0.074 |
| Negative | 3 | 22 | |
| Nottingham histological grade | | | |
| 1+2 | 5 | 21 | 0.386 |
| 3 | 11 | 25 | |
| Ki-67 labeling index (LI) | | | |
| High | 9 | 27 | 1.00 |
| Low | 6 | 16 | |
| Not tested | 1 | 3 | |
| Stromal tumor-infiltrating lymphocytes | | | |
| High | 0 | 19 | *0.001* |
| Low/intermediate | 16 | 27 | |
| Adjuvant chemotherapy | | | |
| Performed | 7 | 27 | 0.549 |
| Not performed | 7 | 18 | |
| Undetermined | 2 | 1 | |

patients. FF was significantly associated with venous invasion and a high Nottingham histological grade ($p = 0.003$ and $0.033$, respectively). However, pathological stage, lymph node status, and lymphatic invasion were not significantly different across both groups. Presence of FF was significantly associated with low/intermediate TILs in the stroma ($p = 0.001$). Fig 2D shows the OS and RFS curves of FF-positive and -negative patients. Presence of FF was significantly correlated with poor OS ($p = 0.012$) and RFS ($p = 0.028$) in patients with TNBC.

## Association of combined myxoid change and FF with clinicopathological parameters

Both myxoid change and FF were observed simultaneously in 11 patients (17.7%), and only myxoid change or FF were noted in 5 and 10 patients, respectively. Fig 3 shows the OS and

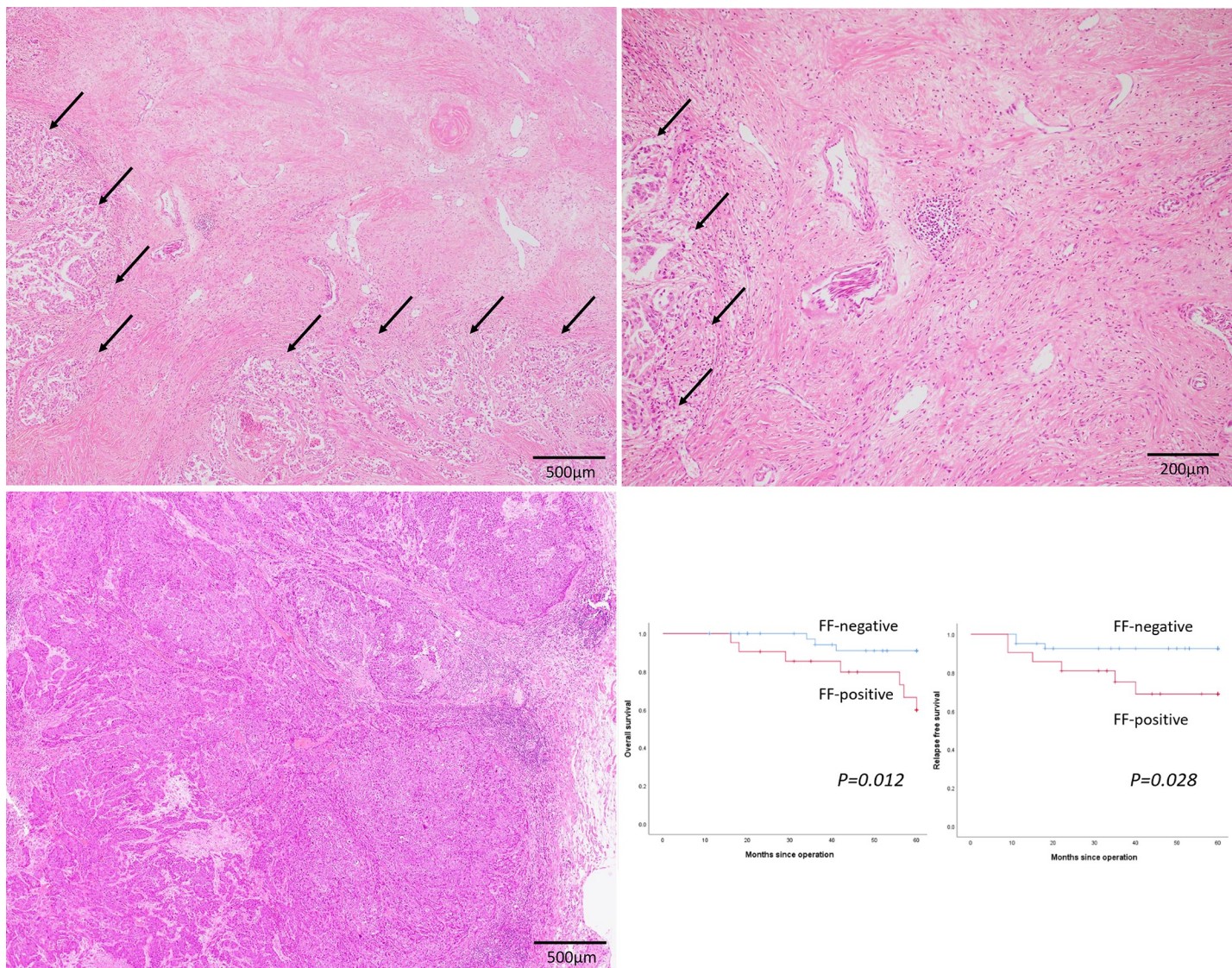

**Fig 2. Fibrotic focus as an independent predictor of survival in patients with triple-negative breast cancer.** (a) (b) Representative image of fibrotic focus (FF) in triple-negative breast cancer. Fibrotic area within the tumor containing collagen surrounded by a highly cellular zone of infiltrating carcinoma (arrows) (x 40 (a) and x 100 (b)). (c) No FF is observed within the tumor (x 40). (d) Kaplan-Meier curves for the overall survival (OS) and relapse-free survival (RFS) of triple-negative patients with breast cancer. OS (left) and RFS (right) curves among FF-positive (red line) and FF-negative (blue line) patients.

RFS curves of myxoid change- and FF-positive vs. myxoid change- and/or FF-negative patients. Presence of both myxoid change and FF was significantly correlated with poor OS (*p = 0.002*) and RFS (*p = 0.001*) in patients with TNBC.

Table 4 summarizes the association of both myxoid change- and FF-positive patients vs. myxoid change- and/or FF-negative patients with clinicopathological factors. Myxoid change- and FF-positive patients were significantly associated with the presence of venous invasion and a high Nottingham histological grade (*p = 0.002* and *0.044*, respectively) (Table 4). However, pathological stage, lymph node status, lymphatic invasion, and Ki-67 LI were not significantly different between both groups. Presence of myxoid change and FF were significantly associated with low/intermediate TILs in the stroma (*p = 0.013*).

**Table 3. Association of fibrotic focus with clinicopathological factors.**

| Factors | Fibrotic focus -positive (n = 21) | Fibrotic focus -negative (n = 41) | *P*-value |
|---|---|---|---|
| Age (years; median ± standard deviation) | 74±16 | 67±14 | 0.067 |
| Menopausal status | | | |
| Premenopausal | 3 | 6 | 1.00 |
| Postmenopausal | 18 | 34 | |
| Unknown | 0 | 1 | |
| Body mass index | | | |
| | 23.4±3.6 | 23.0±3.7 | 0.800 |
| Tumor size (mm) | | | |
| ≦20 | 7 | 25 | 0.060 |
| >20 | 14 | 16 | |
| Pathological stage | | | |
| I + II | 17 | 37 | 0.426 |
| III | 4 | 4 | |
| Lymph node status | | | |
| Positive | 7 | 7 | 0.080 |
| Negative | 7 | 26 | |
| Not tested | 7 | 8 | |
| Lymphatic invasion | | | |
| Positive | 20 | 33 | 0.150 |
| Negative | 1 | 8 | |
| Venous invasion | | | |
| Positive | 18 | 19 | *0.003* |
| Negative | 3 | 22 | |
| Nottingham histological grade | | | |
| 1+2 | 6 | 24 | *0.033* |
| 3 | 15 | 17 | |
| Ki-67 labeling index (LI) | | | |
| High | 13 | 23 | 0.783 |
| Low | 7 | 15 | |
| Not tested | 1 | 3 | |
| Stromal tumor-infiltrating lymphocytes | | | |
| High | 0 | 19 | *0.001* |
| Low/intermediate | 21 | 22 | |
| Adjuvant chemotherapy | | | |
| Performed | 8 | 26 | 0.158 |
| Not performed | 11 | 14 | |
| Undetermined | 2 | 1 | |

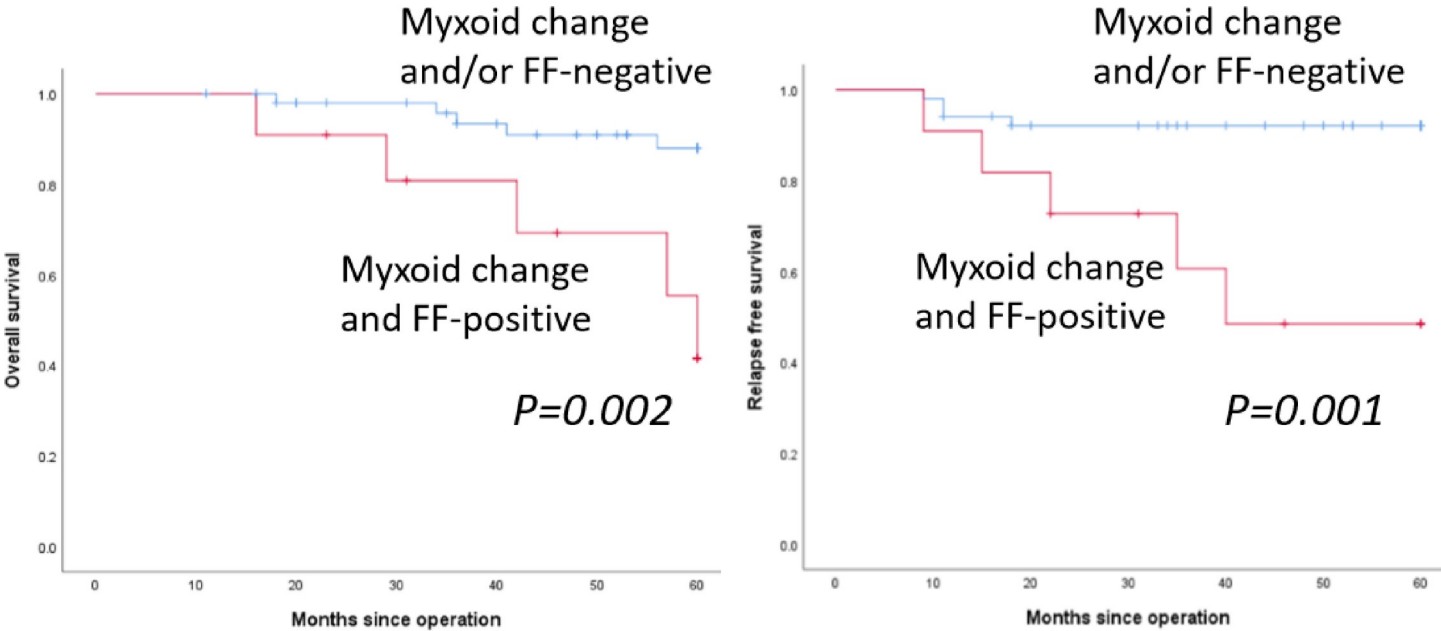

**Fig 3. Survival for myxoid change- and fibrotic focus-positive patients vs. myxoid change- and/or fibrotic focus-negative patients.** Overall survival (OS) (left) and relapse-free survival (RFS) (right) curves in myxoid change and FF-positive (red line) and myxoid change- and/or FF-negative (blue line) patients.

Moreover, a multivariate analysis of RFS clearly demonstrated that the simultaneous presence of myxoid change and FF was an independent poor prognostic factor (hazard ratio [HR] 11.61; 95% confidence interval [CI] 1.027–131.2; $p = 0.048$) (Table 5). In addition, a multivariate analysis of RFS, performed as a separate factor of myxoid change or FF showed that both myxoid change (HR 1.78; 95% CI 0.13–23.7; $p = 0.66$) and FF (HR 5.18; 95% CI 0.34–79.7; $p = 0.24$) were not independent poor prognostic factors.

## Association between myxoid change, FF, and a combination of myxoid change and FF and survival in patients with adjuvant chemotherapy

Consequently, we performed survival analyses in patients with or without adjuvant chemotherapy. Myxoid change was observed in 7 of 34 patients (20.6%) with adjuvant chemotherapy. Fig 4A shows the OS and RFS curves of myxoid change-positive and -negative patients with adjuvant chemotherapy. Presence of myxoid change was significantly correlated with poor RFS ($p = 0.002$), but not with OS ($p = 0.080$). FF was observed in 8 of 34 patients (23.5%) with adjuvant chemotherapy. Fig 4B shows the OS and RFS curves of FF-positive and -negative patients with adjuvant chemotherapy. Presence of FF was significantly correlated with poor RFS ($p = 0.004$), but not with OS ($p = 0.13$). Both myxoid change and FF were observed in 4 of 34 patients (11.8%) with adjuvant chemotherapy. Fig 4C shows the OS and RFS curves of myxoid change- and FF-positive vs myxoid change- and/or FF-negative patients with adjuvant chemotherapy. The presence of a combination of myxoid change and FF was significantly correlated with poor OS ($p = 0.007$) and RFS ($p<0.001$).

## Association between myxoid change, FF, and a combination of myxoid change and FF and survival in patients without adjuvant chemotherapy

Myxoid change was observed in 7 of 25 patients (28.0%) without adjuvant chemotherapy. Fig 5A shows the OS and RFS curves of myxoid change-positive and -negative patients without

**Table 4. Association of myxoid change and fibrotic focus with clinicopathological factors.**

| Factors | Myxoid change and fibrotic focus-positive (n = 11) | Myxoid change and/or fibrotic focus-negative (n = 51) | *P*-value |
|---|---|---|---|
| Age (years old; median ± standard deviation) | 64±19 | 68±14 | 0.658 |
| Menopausal status | | | |
| Premenopausal | 2 | 7 | 0.660 |
| Postmenopausal | 9 | 43 | |
| Unknown | 0 | 1 | |
| Body mass index | | | |
| | 21.9±3.7 | 23.3±3.6 | 0.246 |
| Tumor size (mm) | | | |
| ≦20 | 3 | 28 | 0.182 |
| >20 | 8 | 23 | |
| Pathological stage | | | |
| I + II | 8 | 46 | 0.142 |
| III | 3 | 5 | |
| Lymph node status | | | |
| Positive | 3 | 11 | 0.344 |
| Negative | 3 | 30 | |
| Not tested | 5 | 10 | |
| Lymphatic invasion | | | |
| Positive | 11 | 42 | 0.348 |
| Negative | 0 | 9 | |
| Venous invasion | | | |
| Positive | 11 | 26 | *0.002* |
| Negative | 0 | 25 | |
| Nottingham histological grade | | | |
| 1+2 | 2 | 28 | *0.044* |
| 3 | 9 | 23 | |
| Ki-67 labeling index (LI) | | | |
| High | 4 | 25 | 0.730 |
| Low | 6 | 23 | |
| Not tested | 1 | 3 | |
| Stromal tumor-infiltrating lymphocytes | | | |
| High | 0 | 19 | *0.013* |
| Low/intermediate | 11 | 32 | |
| Adjuvant chemotherapy | | | |
| Performed | 4 | 30 | 0.297 |
| Not performed | 6 | 19 | |
| Undetermined | 1 | 2 | |

adjuvant chemotherapy. Presence of myxoid change was significantly correlated with poor OS (*p = 0.030*), but not with poor RFS (*p = 0.069*). FF was observed in 11 of 25 patients (44.0%) without adjuvant chemotherapy. Fig 5B shows the OS and RFS curves of FF-positive and -negative patients without chemotherapy. Presence of FF was not significantly correlated with OS (*p = 0.068*) and RFS (*p = 0.080*). Both myxoid change and FF were observed in 6 of 25 patients (24.0%) without adjuvant chemotherapy. Fig 5C shows the OS and RFS curves of myxoid change- and FF-positive vs myxoid change- and/or FF-negative patients without adjuvant chemotherapy. The presence of a combination of myxoid change and FF was significantly correlated with poor OS (*p = 0.027*), but not with RFS (*p = 0.68*).

**Table 5. Multivariate analysis of relapse-free survival.**

| Factor | Multivariate analysis | | |
|---|---|---|---|
| | HR | 95% CI | *P*-value |
| Tumor size (mm) | | | |
| 20 < vs. ≦ 20 | 0.297 | 0.021–4.093 | 0.364 |
| Lymph node status | | | |
| Positive vs. negative | 11.82 | 0.647–216.0 | 0.096 |
| Lymphatic invasion | | | |
| Positive vs. negative | $4.846 \times 10^5$ | 0 | 0.980 |
| Venous invasion | | | |
| Positive vs. negative | 0.560 | 0.016–19.50 | 0.749 |
| Nottingham histological grade | | | |
| 3 vs. 1+2 | 1.721 | 0.109–27.16 | 0.700 |
| Ki-67 labelling index (LI) | | | |
| High vs. low/intermediate | 2.240 | 0.163–30.86 | 0.547 |
| Stromal tumor-infiltrating lymphocytes | | | |
| High vs. low | 2.775 | 0.186–41.43 | 0.459 |
| Adjuvant chemotherapy | | | |
| Not performed vs. performed | 5.462 | 0.519–57.52 | 0.158 |
| Myxoid change and fibrotic focus | | | |
| Positive vs. negative | 11.61 | 1.027–131.2 | ***0.048*** |

## Comparison of the presence of myxoid change between pre-operative biopsy specimens and surgically resected specimens

Pre-operative biopsy specimens were available in 44 patients (71%) in this cohort. Of the 11 myxoid change-positive patients diagnosed via surgically resected specimens, 9 patients were positive for myxoid change via biopsy specimens. Besides, of the 33 myxoid change-negative patients diagnosed via surgically resected specimens, 25 patients were negative for myxoid change via the biopsy specimens. Regarding the presence of myxoid change, there was a moderate correlation between biopsy and surgically resected specimens (Kappa statistics: 0.487, *p = 0.001*). FF was noted only in 5 biopsy specimens; thus, statistical analysis was not performed.

## Discussion

TNBC is one of the most challenging molecular subtypes of invasive breast cancer because of its aggressive clinical behavior and the scarcity of targeted therapies; however, recently, an anti-programmed cell death 1 (anti-PD-L1) targeted therapy has emerged as a promising treatment for TNBC [15]. There are no established prognostic factors in patients with TNBC, although some histopathological prognostic indicators have been proposed. FF is a scar-like area in the center of a carcinoma, representing hypoxia [10]. One report demonstrated that FF was associated with higher histological grades, higher T stage, frequent lymph node metastasis, and poor prognosis [16]. In contrast, another study concluded that FF was found in as many as 18% of patients with breast cancer (including all molecular subtypes), and hence was not a significant prognostic factor [4]. In the present cohort, the presence of FF was a significant poor prognostic indicator for both OS and RFS, which is consistent with the results of the first study [10, 16].

Stromal reaction has also attracted attention in various types of carcinomas [6, 7] because the tumor microenvironment plays important roles in cancer growth, invasion, and metastasis.

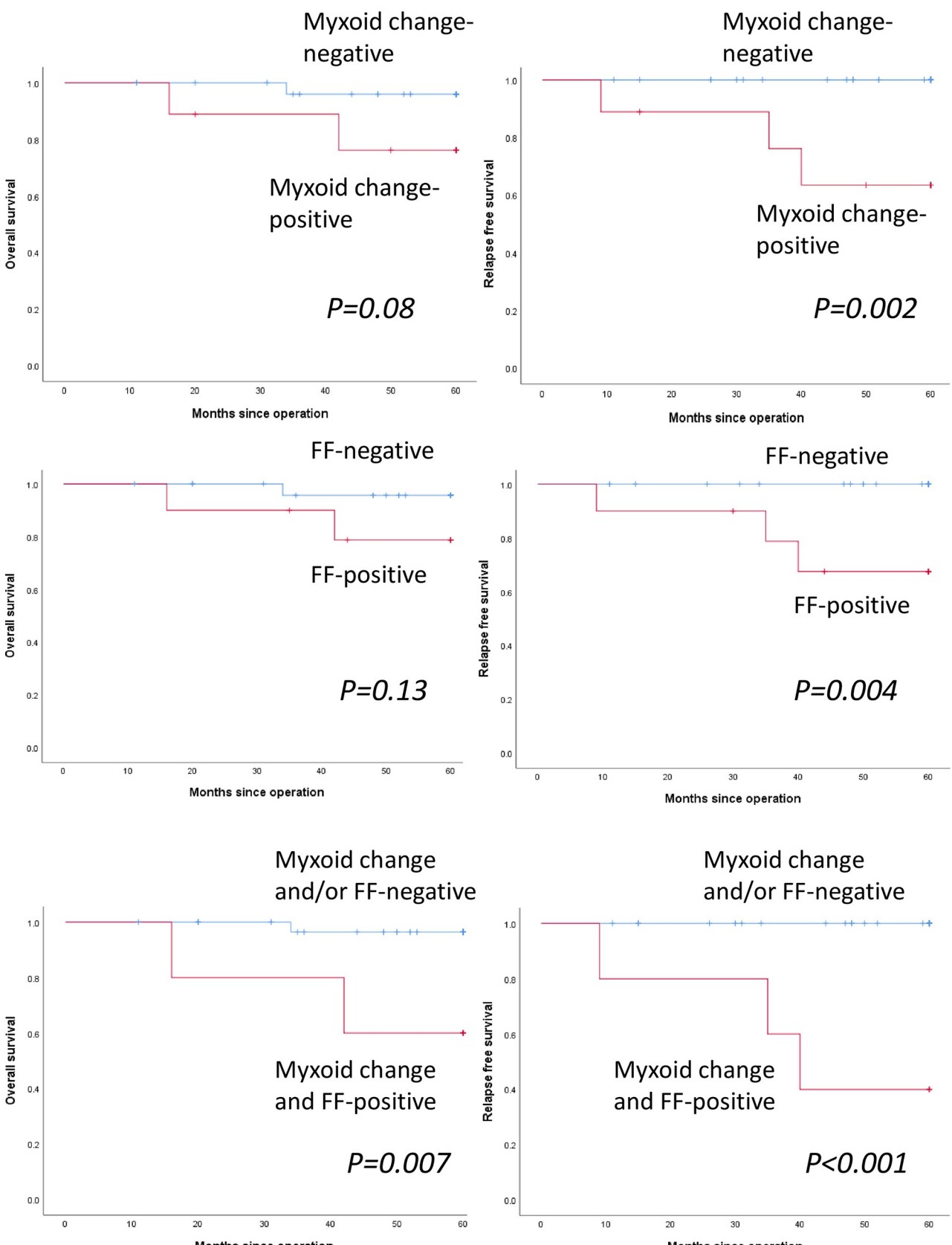

**Fig 4. Survival analyses for myxoid change, fibrotic focus (FF), and a combination of myxoid change and FF in patients with adjuvant chemotherapy.** (a) Kaplan–Meier curves for the overall survival (OS) and relapse-free survival (RFS) of patients with adjuvant chemotherapy. OS (left) and RFS (right) curves among myxoid change-positive (red line) and myxoid change-negative (blue line) patients. (b) OS (left) and RFS (right) curves among FF-positive (red line) and FF-negative (blue line) patients. (c) Overall survival (OS) (left) and relapse-free survival (RFS) (right) curves in myxoid change and FF-positive (red line) and myxoid change- and/or FF-negative (blue line) patients.

Further, stromal reaction is one of the main components of the tumor microenvironment [1]. However, stromal reaction in breast cancer has not received enough attention, although the tumor–stroma ratio (or percentage) has been shown to have a prognostic value [17, 18]. In addition, it has been reported that stromal myxoid change represented hyaluronan accumulation and was significantly associated with lymph node metastasis and a higher histological tumor grade; however, it was not an independent prognostic indicator in patients with breast cancer (molecular subtypes were not evaluated) [19]. Moreover, a recent report showed that 26.9% of the breast cancer specimens, which included all molecular subtypes, had myxoid (immature) stroma, and patients with myxoid stroma showed a significantly poorer survival [8]. Myxoid stroma in breast cancer was linked to a higher histological grade and positive lymph nodes [20]. Therefore, the presence of myxoid stroma may be a poor prognostic indicator in breast cancer, consistent with the findings in colorectal cancer [6, 7]. However, the prognostic significance of the presence of myxoid stroma in patients with TNBC had not been explored. In the present study, we demonstrated that myxoid change was a significant poor prognostic indicator for OS and RFS in patients with TNBC. Interestingly, we further revealed that the simultaneous presence of myxoid change and FF was significantly correlated with poorer OS and RFS, compared with the presence of either myxoid change or FF. To the best of our knowledge, this is the first study, through multivariate analysis, to show that a combination of myxoid change and FF was an independent poor prognostic indicator in patients with TNBC. The results of the present study indicate that histopathological assessment of myxoid change and FF in breast cancer tissues of patients with TNBC may be a useful, practical, and easily assessable method for predicting such patients' prognosis. Although the present study demonstrated a moderate correlation between presence of myxoid change and both pre-operative biopsy and surgically resected specimens, the prognostic significance of presence of myxoid change in the pre-operative biopsy specimens must be clarified in a larger cohort.

Moreover, for the first time, the present study analyzed the survival significance of the presence of myxoid change and/or FF in patients with TNBC receiving adjuvant chemotherapy or not. According to the results of the present study, the presence of myxoid change, FF, and the combination of myxoid change and FF were significant poor prognostic factors for RFS in patients with adjuvant chemotherapy. However, this trend was not noted in patients without adjuvant chemotherapy. Besides, myxoid change-, FF-, and myxoid change and FF-positive patients showed similar survival curves regardless of the presence of adjuvant chemotherapy; however, myxoid change-, FF-, and myxoid change and/or FF-negative patients without adjuvant chemotherapy showed poorer survival curves compared to those of patients with adjuvant chemotherapy. Accordingly, adjuvant chemotherapy may not be efficient in improving outcomes in myxoid change- and/or FF-positive patients with TNBC.

However, the molecular events explaining the association of myxoid change with poor prognosis remain unknown, and this must be revealed to establish novel targeted therapies to treat patients with myxoid stroma who show a poor survival. Notably, there have been reports that the possible hypothesis explaining molecular events in myxoid change in colorectal cancer was associated with inhibition of angiogenesis, which hinders access of the cancer cells by immune cells, leading to altered effects of cancer-associated fibroblasts on the local tumor microenvironment [6]. Notably, in the present cohort, the combination of myxoid change and

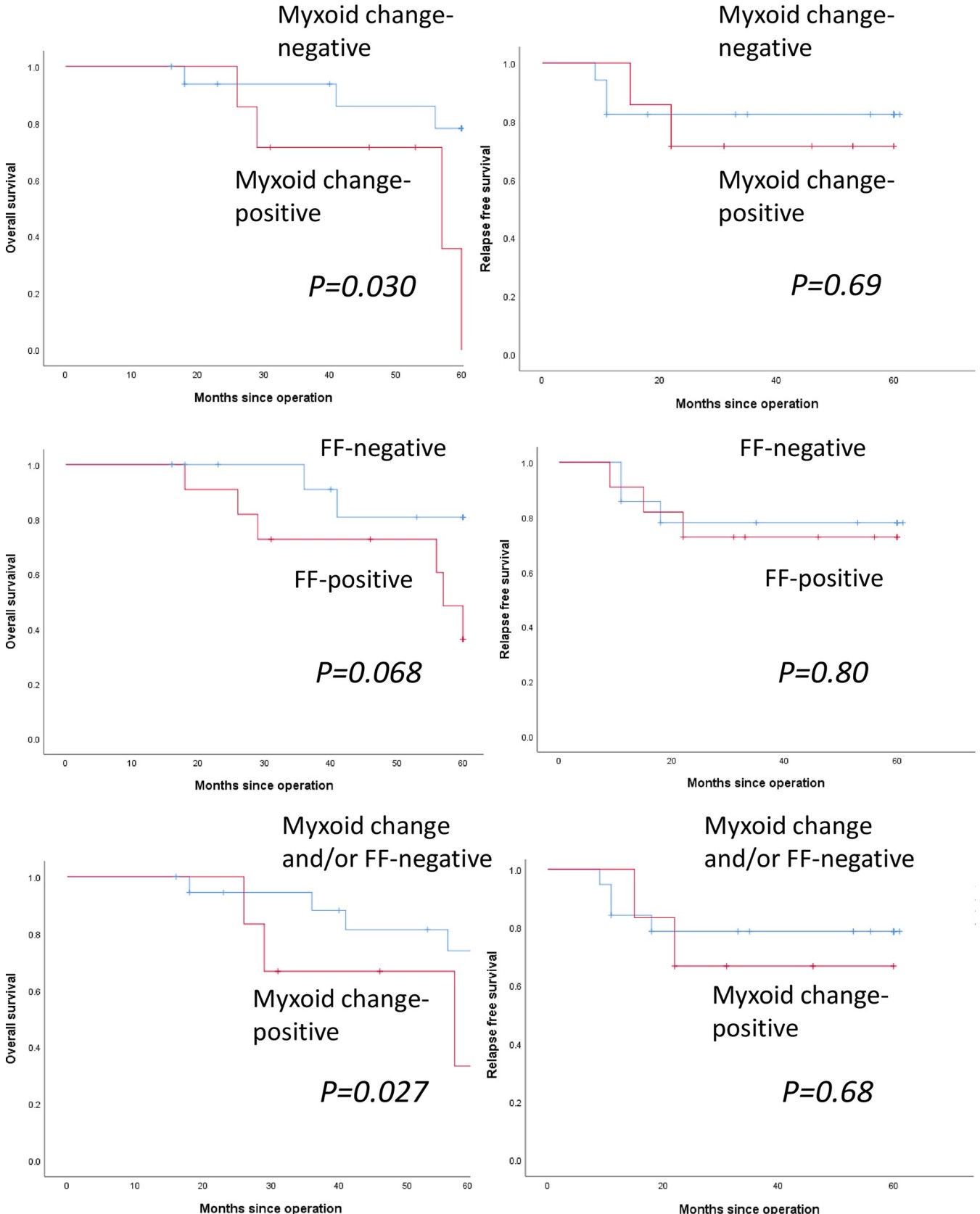

**Fig 5. Survival analyses for myxoid change, fibrotic focus (FF), and a combination of myxoid change and fibrotic focus in patients without adjuvant chemotherapy.** (a) Kaplan–Meier curves for the overall survival (OS) and relapse-free survival (RFS) of patients without adjuvant chemotherapy. OS (left) and RFS (right) curves among myxoid change-positive (red line) and myxoid change-negative (blue line) patients. (b) OS (left) and RFS (right) curves among FF-positive (red line) and FF-negative (blue line) patients. (c) Overall survival (OS) (left) and relapse-free survival (RFS) (right) curves in myxoid change and FF-positive (red line) and myxoid change- and/or FF-negative (blue line) patients.

FF was significantly associated with low/intermediate TILs. Moreover, the association between TILs and desmoplasia in metastatic breast cancer has also been reported [21]. Therefore, the presence of myxoid change and FF in TNBC stroma may be a barrier to immune-cell infiltration. Furthermore, the detailed mechanism of myxoid change in the tumor stroma, which represents hyaluronan accumulation, is unclear, although the patterns of stromal reaction are recognized to be associated with tumor microenvironment, including immune cells and cancer-associated fibroblasts [6]. Neoadjuvant chemotherapy may affect the tumor microenvironment, which may lead to a change of the stromal reaction of TNBC. Thus, additional studies are needed to clarify the molecular events underlying the formation of myxoid change in TNBC stroma.

There are some limitations in the present study. First, the present retrospective study included a relatively small number of patients with TNBC. Therefore, additional validation studies with better designs are needed to clarify the prognostic significance of myxoid change and FF in patients with TNBC. Second, definitive assessable and reproductive criteria for myxoid change and FF are needed. Therefore, we performed this study using the previously described criteria of myxoid change that were used in colorectal cancer [6, 7] and FF [10]. Diagnostic criteria of myxoid change and FF for breast cancer specimens must be established for routine use to predict the prognosis.

## Conclusions

This study clearly suggests that myxoid change and FF are independent poor prognostic indicators in patients with TNBC of the present cohort. Additionally, the combination of myxoid change and FF is also an independent poor prognostic indicator in patients with TNBC, as seen in the multivariate analysis in this study. Besides, our findings suggest that the presence of myxoid change and FF in TNBC stroma may be a barrier to immune-cell infiltration. The results of the present study indicate that histopathological assessment of myxoid change and FF in breast cancer tissues of patients with TNBC may be a useful, practical, and easily assessable method for predicting the prognosis of patients with TNBC, which should be confirmed in prospective studies with larger samples. In addition, diagnostic criteria for the establishment of myxoid change and FF in TNBC must be established. Additional studies are also required to clarify the molecular events underlying the myxoid change in TNBC.

## Supporting information

**S1 Dataset.**
(PDF)

## Author Contributions

**Conceptualization:** Hirotsugu Yanai.

**Data curation:** Hirotsugu Yanai.

**Formal analysis:** Hirotsugu Yanai, Katsuhiro Yoshikawa.

**Investigation:** Hirotsugu Yanai.

**Methodology:** Hirotsugu Yanai, Katsuhiro Yoshikawa, Mitsuaki Ishida.

**Project administration:** Hirotsugu Yanai, Mitsuaki Ishida.

**Supervision:** Koji Tsuta, Mitsugu Sekimoto, Tomoharu Sugie.

**Validation:** Mitsuaki Ishida.

**Writing – original draft:** Hirotsugu Yanai, Mitsuaki Ishida.

**Writing – review & editing:** Koji Tsuta, Mitsugu Sekimoto, Tomoharu Sugie.

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
