## [Decision Letter · Decision Letter 0]

23 Sep 2020

PONE-D-20-23411

Presence of myxoid stromal change and fibrotic focus in pathological examination are prognostic factors of triple-negative breast cancer: Results from a retrospective single-center study

PLOS ONE

Dear Dr. Ishida,

Thank you for submitting your manuscript to PLOS ONE. After careful consideration, we feel that it has merit but does not fully meet PLOS ONE’s publication criteria as it currently stands. Therefore, we invite you to submit a revised version of the manuscript that addresses the points raised during the review process.

We look forward to receiving your revised manuscript.

Kind regards,

Elda Tagliabue

Academic Editor

PLOS ONE

Journal Requirements:

2.In your Data Availability statement, you have not specified where the minimal data set underlying the results described in your manuscript can be found. PLOS defines a study's minimal data set as the underlying data used to reach the conclusions drawn in the manuscript and any additional data required to replicate the reported study findings in their entirety. All PLOS journals require that the minimal data set be made fully available. For more information about our data policy, please see http://journals.plos.org/plosone/s/data-availability.

3.Your ethics statement should only appear in the Methods section of your manuscript. If your ethics statement is written in any section besides the Methods, please delete it from any other section.

Reviewers' comments:

Reviewer's Responses to Questions

**Comments to the Author**

1. Is the manuscript technically sound, and do the data support the conclusions?

Reviewer #1: Partly

Reviewer #2: Partly

2. Has the statistical analysis been performed appropriately and rigorously? 

Reviewer #1: Yes

Reviewer #2: No

3. Have the authors made all data underlying the findings in their manuscript fully available?

Reviewer #1: Yes

Reviewer #2: Yes

4. Is the manuscript presented in an intelligible fashion and written in standard English?

Reviewer #1: Yes

Reviewer #2: Yes

5. Review Comments to the Author

Reviewer #1: In this paper the authors studied stromal reaction in TNBC tumors as prognostic factors. Specifically they assessed that the presence of myxoid change and fibrotic focus are poor prognostic factors in TNBC patients in term of overall survival and disease free survival and that the combination of both parameters was an independent poor prognostic factor.

The major limitation of this study lies on the difficulty to generalize the clinical value of the prognostic factors described due to the fact that the authors analyzed only a cohort of 62 TNBC patients without a validation cohort.

The paper is potentially interesting anyway a number of compulsory aspects need to be clarified and the following comments should be addressed before the manuscript can be considered for publication:

Major points:

• The authors analyzed only a cohort of 62 TNBC patients. Please add a validation cohort.

• As reported in Table 1, 56.5% of patients received adjuvant chemotherapy and 38.7% did not received chemotherapy. The analysis of OS and RFS for all stromal prognostic factors considered should be performed considering separately patients who received/not received treatment. Please add these analyses.

• In figures 1a and 2a the authors report only a representative image at large magnification for myxoid change and fibrotic focus respectively. Please add also images at lower magnification and images of tumor sample negative for myxoid change and fibrotic focus.

• Correlation analysis with clinicopathological factors, reported in Table 2 was done only considering patients with both mixoid change and fibrotic focus together. Please add correlation analysis considering separately the two stromal characteristics.

Minor points:

• In the introduction section the authors should better support sentences at lines 62-65 with references from literature

• Table 1: tumor size is considered as a continuous variable. Could the authors indicate tumor size as they do in Table 2 and 3 for correlation and multivariate analyzes?

• As regards the analysis of TILs, the paper the authors refer to, considers also “intermediate TILs as 10%<tils≤59%”. analysis="">

Overall the manuscript can be interesting but it cannot be accepted in the present form. A major revision is mandatory before publication.</tils≤59%”.>

Reviewer #2: Hirotsugu Yanai and collaborators examined on 62 triple negative breast cancers the clinical meaning of the histopathological assessment of myxoid change and FF fibrotic focus. Although their results seem to be novel and promising there are important issues that need to be addressing.

Major issues

Methods

1. Although there are two cited papers regarding the criteria to detect myxoid change it would be useful to clearly explain the characteristics taking into account for the analysis.

2. There is any score to classify positive and negative tissues (%, number of cells, area)?, what was to cutoff to consider a positive tissue?

Results

1. Please include at least two more representative pictures of myxoid changes and FF, and clearly indicate by arrows the area and changes of interest. Please also include images from patients in which the myxoid changes or FF were not observed.

2. On table 2 why positive patients to FF or Myxoid change were grouped with negative patients?. I would group the tumors as following to get a better idea of the independent and coordinate changes: 1) Myxoid change and fibrotic focus-positive (N=11) 2) Myxoid change positive-only (N=5) 3) fibrotic focus-positive (N=10) and 4) Myxoid change and fibrotic focus-positive negative (N=36).

3. As previously mentioned on table 3 it is necessary to show multivariate analysis of stroma changes occurring independently and in co-occurrence, to get an idea of how this phenomena impact tumor biology alone or in combination.

4. Although this is one of the first efforts to associate the presence myxoid changes and FF with clinical outcomes and patients features on TNBC, unfortunately by the limited number of included tumors there is not an optimal statistical strategy that can be robustly applied and the conclusion are not strongly supported, thus I encourage the authors to include some independent cohorts to confirm some of their findings. Since hematoxylin and eosin is the only necessary test to evaluate stroma reactions, I will suggest including some triple negative TCGA tumors that can be evaluated from https://cancer.digitalslidearchive.org/#!/CDSA/brca and clinical and TILS information is available.

5. Since most of the triple negative tumors are now being treated in neoadjuvant therapy, it is possible to examine this stroma reactions from biopsy tissues?. The authors described that they discard an important set of tumoral samples from their original cohort due to neoadjuvant treatment. So 1) It is possible to compare the histopathological assessment of the surgical analyzed tissues, included in this study, with their matched diagnosis biopsies to define how much a biopsy tissue can recapitulate what is detected in a larger tissue (surgical sample) 2) it is possible to include the analysis of this excluded patients based on their biopsy tissue? Please discuss this point.

Minor

Since a Fisher test was computed to examine the significance of the association between two kinds of classification the most accurate term for table 2 and methods section is association instead of correlation.

6. PLOS authors have the option to publish the peer review history of their article (what does this mean?). If published, this will include your full peer review and any attached files.

Reviewer #1: No

Reviewer #2: **Yes: **Sandra Romero-Cordoba

---

## [Author Response · Author response to Decision Letter 0]

3 Dec 2020

November 30, 2020

Dr. Joerg Heber

Editor-in-Chief

PLoS ONE

Resubmission - Manuscript ID: PONE-D-20-23411

Dear Dr. Heber:

I would like to resubmit an article for publication in PLoS ONE, titled “Presence of myxoid stromal change and fibrotic focus in pathological examination are prognostic factors of triple-negative breast cancer: Results from a retrospective single-center study.” 

Your comments, as well as those of the reviewers, were highly insightful and enabled us to greatly improve the quality of our manuscript. In the following pages, I have provided our point-by-point response to all comments and have highlighted the revised portions of the manuscript.

I hope that you will find our revised manuscript suitable for publication in PLoS ONE. I look forward to hearing from you at your earliest convenience.

Sincerely,

Mitsuaki Ishida 

Kansai Medical University 

2-5-1, Shinmachi, Hirakata City

Osaka, 573-1010, Japan

Tel: +81-72-804-2794

Fax: +81-72-804-2794

E-mail: ishidamt@hirakata.kmu.ac.jp

Response to the comments of Reviewer #1

Thank you very much for reviewing our manuscript. We appreciate your constructive comments. We have made the following revisions in response to the issues raised by you.

1. The authors analyzed only a cohort of 62 TNBC patients. Please add a validation cohort.

Response: This study cohort included a relatively small number of patients with triple-negative breast cancer (TNBC), as mentioned in the Discussion section. As you have mentioned, a validation cohort is required to clarify the results of the present study. However, at this point, we are unable to validate the results, and we aim to undertake an international validation cohort study to clarify the usefulness of myxoid change and fibrotic focus in TNBC in the near feature. 

2. As reported in Table 1, 56.5% of patients received adjuvant chemotherapy and 38.7% did not received chemotherapy. The analysis of OS and RFS for all stromal prognostic factors considered should be performed considering separately patients who received/not received treatment. Please add these analyses.

Response: As you have suggested, we have additional analyses of OS and RFS in TNBC patients with or without adjuvant chemotherapy. We added the results of these analyses in the Results section and showed the survival curves in Figs. 4 and 5 (p19-22). Moreover, we added the following statement regarding the results in the Discussion section. 

“Association between myxoid change, FF, and a combination of myxoid change and FF and survival in patients with adjuvant chemotherapy

 Consequently, we performed survival analyses in patients with or without adjuvant chemotherapy. Myxoid change was observed in 7 of 34 patients (20.6%) with adjuvant chemotherapy. Fig. 4a shows the OS and RFS curves of myxoid change-positive and -negative patients with adjuvant chemotherapy. The presence of myxoid change significantly correlated with poor RFS (p=0.002) but not with OS (p=0.080). FF was observed in 8 of 34 patients (23.5%) with adjuvant chemotherapy. Fig. 4b shows the OS and RFS curves of FF-positive and -negative patients with adjuvant chemotherapy. The presence of FF significantly correlated with poor RFS (p=0.004) but not with OS (p=0.13). Both myxoid change and FF were observed in 4 of 34 patients (11.8%) with adjuvant chemotherapy. Fig. 4c shows the OS and RFS curves of myxoid change- and FF-positive vs myxoid change-and/or FF-negative patients with adjuvant chemotherapy. The presence of a combination of myxoid change and FF significantly correlated with poor OS (p=0.007) and RFS (p<0.001).

Fig. 4 Survival analyses for myxoid change, fibrotic focus (FF), and a combination of myxoid change and FF in patients with adjuvant chemotherapy. 

(a) Kaplan–Meier curves for the overall survival (OS) and relapse-free survival (RFS) of patients with adjuvant chemotherapy. OS (left) and RFS (right) curves among myxoid change-positive (red line) and myxoid change-negative (blue line) patients. (b) OS (left) and RFS (right) curves among FF-positive (red line) and FF-negative (blue line) patients. (c) Overall survival (OS) (left) and relapse-free survival (RFS) (right) curves in myxoid change and FF-positive (red line) and myxoid change- and/or FF-negative (blue line) patients. 

Association between myxoid change, FF, and a combination of myxoid change and FF and survival in patients without adjuvant chemotherapy

 Myxoid change was observed in 7 of 25 patients (28.0%) without adjuvant chemotherapy. Fig. 5a shows the OS and RFS curves of myxoid change-positive and -negative patients without adjuvant chemotherapy. The presence of myxoid change significantly correlated with poor OS (p=0.030) but not with poor RFS (p=0.069). FF was observed in 11 of 25 patients (44.0%) without adjuvant chemotherapy. Fig. 5b shows the OS and RFS curves of FF-positive and -negative patients without chemotherapy. The presence of FF did not significantly correlate with OS (p=0.068) and FRS (p=0.080). Both myxoid change and FF were observed in 6 of 25 patients (24.0%) without adjuvant chemotherapy. Fig. 5c shows the OS and RFS curves of myxoid change- and FF-positive vs myxoid change-and/or FF-negative patients without adjuvant chemotherapy. The presence of a combination of myxoid change and FF significantly correlated with poor OS (p=0.027) but not with RFS (p=0.68).

Fig. 5 Survival analyses for myxoid change, fibrotic focus (FF), and a combination of myxoid change and fibrotic focus in patients without adjuvant chemotherapy. 

(a) Kaplan–Meier curves for the overall survival (OS) and relapse-free survival (RFS) of patients without adjuvant chemotherapy. OS (left) and RFS (right) curves among myxoid change-positive (red line) and myxoid change-negative (blue line) patients. (b) OS (left) and RFS (right) curves among FF-positive (red line) and FF-negative (blue line) patients. (c) Overall survival (OS) (left) and relapse-free survival (RFS) (right) curves in myxoid change and FF-positive (red line) and myxoid change- and/or FF-negative (blue line) patients.” (page 19-22)

“Moreover, for the first time, the present study analyzed the survival significance of the presence of myxoid change and/or FF in TNBC patients with or without adjuvant chemotherapy. According to the results of the present study, the presence of myxoid change, FF, and the combination of myxoid change and FF were significant poor prognostic factors for RFS in patients with adjuvant chemotherapy. However, this trend was not noted in patients without adjuvant chemotherapy. Besides, myxoid change-, FF-, and myxoid change and FF-positive patients showed similar survival curves regardless of the presence of adjuvant chemotherapy; however, myxoid change-, FF-, and myxoid change and/or FF-negative patients without adjuvant chemotherapy showed poorer survival curves compared to those of patients with adjuvant chemotherapy. Accordingly, adjuvant chemotherapy might not be efficient in improving outcomes in myxoid change- and/or FF-positive TNBC patients.” (page 25)

3. In figures 1a and 2a the authors report only a representative image at large magnification for myxoid change and fibrotic focus respectively. Please add also images at lower magnification and images of tumor sample negative for myxoid change and fibrotic focus.

Response: As you have suggested, we have added the images with a lower magnification of myxoid change (Figs. 1a, ab) or fibrotic focus (Figs. 2a, 2b). We have also added the figures without myxoid change (Fig. 1c) or fibrotic focus (Fig. 2c). 

4. Correlation analysis with clinicopathological factors, reported in Table 2 was done only considering patients with both mixoid change and fibrotic focus together. Please add correlation analysis considering separately the two stromal characteristics.

Response: As you have suggested, we have performed additional analyses of the association of myxoid change or fibrotic focus with clinicopathological characteristics, separately. We have illustrated these results in Table 2 and Table 3. The results were fundamentally the same as those of myxoid change and fibrotic focus (Table 4). 

5. In the introduction section the authors should better support sentences at lines 62-65 with references from literature

Response: As you have suggested, we have added reference 5. 

6. Table 1: tumor size is considered as a continuous variable. Could the authors indicate tumor size as they do in Table 2 and 3 for correlation and multivariate analyzes?

Response: We agree with you that tumor size is a continuous variable. In this cohort, the median tumor size was 20 mm. Notably, TNM classification of the breast cancer classified T1 as 2cm or less, and T2 and T3 as more than 2 cm. Thus, we classified the tumor size as ≦2 cm or >2 cm in this analysis. 

7. As regards the analysis of TILs, the paper the authors refer to, considers also “intermediate TILs as 10%

Response: As you have suggested, TIL is classified as low (0-10%), intermediate (11-59%), and high (more than 60%) according to Reference 12. Thus, in this study, we classified TIL as low/intermediate (0-59%) and high (more than 60%). 

Response to the comments of Reviewer #2

Thank you very much for reviewing our manuscript. We appreciate your constructive comments. We have made the following revisions in response to the issues raised by you.

1. Although there are two cited papers regarding the criteria to detect myxoid change it would be useful to clearly explain the characteristics taking into account for the analysis.

Response: As you have suggested, we have added a statement regarding the histopathological characteristics of myxoid change in the methods section. 

“Myxoid change was histopathologically characterized by the presence of amorphous extracellular substances, which comprise an amphophilic or slightly basophilic material within the fibrous stroma” (page 6)

2. There is any score to classify positive and negative tissues (%, number of cells, area)?, what was to cutoff to consider a positive tissue?

Response: In the present study, we classified tissues as positive when typical myxoid change was present around the tumor according to colorectal cancer criteria (Ref. 6,7), and no cutoff value was present. As you have suggested, definitive assessable and reproductive criteria for myxoid change in breast cancer are needed (Discussion, page 26). 

3. Please include at least two more representative pictures of myxoid changes and FF, and clearly indicate by arrows the area and changes of interest. Please also include images from patients in which the myxoid changes or FF were not observed.

Response: As you have suggested, we have added images with a lower magnification of myxoid change (Figs. 1a, ab) or fibrotic focus (Figs. 2a, 2b). We have also added the figures without myxoid change (Fig. 1c) or fibrotic focus (Fig. 2c). 

4. On table 2 why positive patients to FF or Myxoid change were grouped with negative patients?. I would group the tumors as following to get a better idea of the independent and coordinate changes: 1) Myxoid change and fibrotic focus-positive (N=11) 2) Myxoid change positive-only (N=5) 3) fibrotic focus-positive (N=10) and 4) Myxoid change and fibrotic focus-positive negative (N=36).

Response: We have performed additional analyses regarding the association of myxoid change or fibrotic focus with clinicopathological characteristics, separately. We have illustrated these results in Table 2 and Table 3. The results were fundamentally the same as those of myxoid change/fibrotic focus (Table 4). We think that analyses of four groups (myxoid change and/or fibrotic focus) might be difficult because there were only 5 patients in the myxoid change-positive/fibrotic focus-negative group. 

5. As previously mentioned on table 3 it is necessary to show multivariate analysis of stroma changes occurring independently and in co-occurrence, to get an idea of how this phenomena impact tumor biology alone or in combination.

Response: As you have suggested, we have performed multivariate analyses of relapse-free survival of myxoid change or fibrotic focus, separately. We have added the results as follows. 

“In addition, a multivariate analysis of RFS performed as a separate factor of myxoid change or FF showed that both myxoid change (HR 1.78; 95% CI 0.13-23.7; p=0.66) and FF (HR 5.18; 95% CI 0.34-79.7; p=0.24) were not independent poor prognostic factors.” (page 18)

6. Although this is one of the first efforts to associate the presence myxoid changes and FF with clinical outcomes and patients features on TNBC, unfortunately by the limited number of included tumors there is not an optimal statistical strategy that can be robustly applied and the conclusion are not strongly supported, thus I encourage the authors to include some independent cohorts to confirm some of their findings. Since hematoxylin and eosin is the only necessary test to evaluate stroma reactions, I will suggest including some triple negative TCGA tumors that can be evaluated from https://cancer.digitalslidearchive.org/#!/CDSA/brca and clinical and TILS information is available.

Response: We appreciate your kind suggestion. This study cohort included a relatively small number of patients with triple-negative breast cancer (TNBC), as we have acknowledged in the Discussion section. As you mentioned, a validation cohort is required to clarify the results of the present study. However, at this point, we are unable to validate the results, and we aim to undertake an international validation cohort study to clarify the usefulness of myxoid change and fibrotic focus in TNBC in the near feature.

In the present study, the presence of myxoid change or fibrotic focus was examined using surgically resected specimens (more than 2 slices in all cases). While the tool you have suggested is very useful, we believe it is not suitable for this study since only one slice was available in each case. 

7. Since most of the triple negative tumors are now being treated in neoadjuvant therapy, it is possible to examine this stroma reactions from biopsy tissues?. The authors described that they discard an important set of tumoral samples from their original cohort due to neoadjuvant treatment. So 1) It is possible to compare the histopathological assessment of the surgical analyzed tissues, included in this study, with their matched diagnosis biopsies to define how much a biopsy tissue can recapitulate what is detected in a larger tissue (surgical sample) 2) it is possible to include the analysis of this excluded patients based on their biopsy tissue? Please discuss this point.

Response: As you have suggested, we have examined the usefulness of analysis of the matched pre-operative biopsy specimens. We have added the results. 

“Moreover, we also evaluated the presence of myxoid change and FF using the matched pre-operative biopsy specimens.” (page 6-7)

“Comparison of the presence of myxoid change between pre-operative biopsy specimens and surgically-resected specimens

 Pre-operative biopsy specimens were available in 44 patients (71%) in this cohort. Of the 11 myxoid change-positive patients diagnosed via surgically resected specimens, 9 patients were positive for the myxoid change via biopsy specimens. Besides, of the 33 myxoid change-negative patients diagnosed via surgically resected specimens, 25 patients were negative for myxoid change via biopsy specimens. There was a moderate correlation between biopsy and surgically resected specimens regarding the presence of myxoid change (Kappa statistics: 0.487, p=0.001). FF was noted only in 5 biopsy specimens; thus, statistical analysis was not performed.” (page 21-22)

“Additionally, the present study demonstrated a moderate correlation between pre-operative biopsy and surgically resected specimens regarding the presence of myxoid change. Thus, evaluation of the presence of myxoid change in the biopsy specimens might provide prognostic information in patients with TNBC because many patients with TNBC are currently treated with neoadjuvant chemotherapy.” (page 24-25)

8. Since a Fisher test was computed to examine the significance of the association between two kinds of classification the most accurate term for table 2 and methods section is association instead of correlation.

Response: As you have suggested, we have changed “correlation” to “association”. 

“The association between groups was evaluated using Fisher’s exact test for categorical variables” (page 7)

---

## [Decision Letter · Decision Letter 1]

17 Dec 2020

PONE-D-20-23411R1

Presence of myxoid stromal change and fibrotic focus in pathological examination are prognostic factors of triple-negative breast cancer: Results from a retrospective single-center study.

PLOS ONE

Dear Dr. Ishida,

Thank you for submitting your manuscript to PLOS ONE. After careful consideration, we feel that it has merit but does not fully meet PLOS ONE’s publication criteria as it currently stands. Therefore, we invite you to submit a revised version of the manuscript that addresses the points raised during the review process.

We look forward to receiving your revised manuscript.

Kind regards,

Elda Tagliabue

Academic Editor

PLOS ONE

Reviewers' comments:

Reviewer's Responses to Questions

**Comments to the Author**

1. If the authors have adequately addressed your comments raised in a previous round of review and you feel that this manuscript is now acceptable for publication, you may indicate that here to bypass the “Comments to the Author” section, enter your conflict of interest statement in the “Confidential to Editor” section, and submit your "Accept" recommendation.

Reviewer #1: All comments have been addressed

Reviewer #2: All comments have been addressed

2. Is the manuscript technically sound, and do the data support the conclusions?

Reviewer #1: Yes

Reviewer #2: Partly

3. Has the statistical analysis been performed appropriately and rigorously? 

Reviewer #1: Yes

Reviewer #2: Yes

4. Have the authors made all data underlying the findings in their manuscript fully available?

Reviewer #1: Yes

Reviewer #2: Yes

5. Is the manuscript presented in an intelligible fashion and written in standard English?

Reviewer #1: Yes

Reviewer #2: Yes

6. Review Comments to the Author

Reviewer #1: The authors have comprehensively addressed the queries raised in my previous review, except the request to analyze a new cohort to validate results obtained using the cohort of 62 TNBC patients. Anyway the authors have made efforts to enhance the quality of the paper, and this merits consideration.

Only four minor points need to be clarified before the manuscript can be considered for publication.

• Significant results shown in Table 2 and Table 3 need to be reported also in Result section as the authors did for Table 4.

• Figure 2 a, b and c: please use 100X and 200X magnification as done in figure 1. Please add magnification bars in figure 1 and 2.

• “Clearly” line 316 pag 24: since the authors analyzed only a cohort of patients, this adjective should be omitted.

• Sentences lines 337-342 pag 25: since the analysis of prognosis was done considering surgically resected specimens, the conclusion concerning the importance of biopsy should be revised

Reviewer #2: In this revised text Hirotsugu Yanai and collaborators improved their analysis and presented some novel results. However, I continue having the feeling that their small set of evaluated tumors prevent them to conclude some of the statements they are describing throughout the text. Thus, I think this needs to be addressed before publishing.

Point:

1. Since it is not possible to evaluate an independent validation cohort, and given the limited number of tested samples and consequently the poor number of clinical events analyzed, findings from this report should be interpreted with more caution, and thus a mention of this important limitation in the discussion is not enough. Authors should avoid to extend their finding to all TNBC patients, despite, they limited their results to their "cohort" in some sections but not through the entire manuscript as seen by the tittle, and conclusion sections, this need to be addressed.

2. An interesting data is presented by the comparison of myxoid change between pre-operative biopsy specimens and surgically-resected specimens. Can you please add a brief commentary on how this changes are established. How therapeutical interventions can cause this phenomena?

3. Since the "Presence of myxoid change and FF was significantly associated with low/intermediate TILs in the stroma (p=0.013)", what is the advantage to test FF and myxoid change over TILS, which is now a days an standardized pathologic test made on clinical samples, and more robust international guidelines had being described for its evaluation and interpretation. An although some controversial results, this immune measurement has been also related with prognosis and survival outcomes. Please comment this.

7. PLOS authors have the option to publish the peer review history of their article (what does this mean?). If published, this will include your full peer review and any attached files.

Reviewer #1: No

Reviewer #2: No

---

## [Author Response · Author response to Decision Letter 1]

27 Dec 2020

December 27, 2020

Dr. Joerg Heber

Editor-in-Chief

PLoS ONE

Resubmission - Manuscript ID: PONE-D-20-23411R1

Dear Dr. Heber:

I would like to resubmit an article for publication in PLoS ONE, titled “Presence of myxoid stromal change and fibrotic focus in pathological examination are prognostic factors of triple-negative breast cancer: Results from a retrospective single-center study.” 

Your comments, as well as those of the reviewers, were highly insightful and enabled us to greatly improve the quality of our manuscript. In the following pages, we have provided our point-by-point response to all comments and have highlighted the revised portions of the manuscript in red.

I hope that you will find our revised manuscript suitable for publication in PLoS ONE. I look forward to hearing from you at your earliest convenience.

Sincerely,

Mitsuaki Ishida 

Kansai Medical University 

2-5-1, Shinmachi, Hirakata City

Osaka, 573-1010, Japan

Tel: +81-72-804-2794

Fax: +81-72-804-2794

E-mail: ishidamt@hirakata.kmu.ac.jp

Response to the comments of Reviewer #1

Thank you very much for reviewing our manuscript. We appreciate your constructive comments. We have made the following revisions in response to the issues you raised.

1. Significant results shown in Table 2 and Table 3 need to be reported also in Result section as the authors did for Table 4.

Response: As you suggested, we added comments on the significant results shown in Table 2 and Table 3 in the Results section. 

“Myxoid change was significantly associated with low/intermediate TILs in the stroma (p=0.001). However, there were no significant differences in pathological stage, lymph node status, and Nottingham histological grades between the groups.” (page 11, lines 157–160)

“FF was significantly associated with venous invasion and a high Nottingham histological grade (p=0.003 and 0.033, respectively). However, pathological stage, lymph node status, and lymphatic invasion were not significantly different across both groups. Presence of FF was significantly associated with low/intermediate TILs in the stroma (p=0.001).” (page 13, lines 179–183)

2. Figure 2 a, b and c: please use 100X and 200X magnification as done in figure 1. Please add magnification bars in figure 1 and 2.

Response: You suggested that the subpanels of Figure 2 should have higher magnifications as Figure 1. However, we used x 40 for Figures 2a and 2c, and x 100 for Figure 2b because a fibrotic focus is a fibrotic area within the tumor containing fibroblasts and collagen, surrounded by a highly cellular zone of infiltrating carcinoma, and measuring more than 1 mm (as described in page 6-7, lines 106–110). We demonstrated a measure of the fibrotic area within the tumor in Figures 2, thus we did not change the magnifications. 

In addition, we added bars in Figures 1 and 2 as you suggested. 

3. “Clearly” line 316 pag 24: since the authors analyzed only a cohort of patients, this adjective should be omitted.

Response: As you suggested, we omitted the word “clearly”. 

“In the present study, we demonstrated that myxoid change was a significant poor prognostic indicator for OS and RFS in patients with TNBC.” (page 24, lines 322–324)

4. Sentences lines 337-342 pag 25: since the analysis of prognosis was done considering surgically resected specimens, the conclusion concerning the importance of biopsy should be revised

Response: As you suggested, we added a comment on the significance of myxoid change in the pre-operative biopsy specimens. 

“Although the present study demonstrated a moderate correlation between presence of myxoid change and both pre-operative biopsy and surgically resected specimens, the prognostic significance of presence of myxoid change in the pre-operative biopsy specimens must be clarified in a larger cohort.” (page 25, lines 331–335)

Response to the comments of Reviewer #2

Thank you very much for reviewing our manuscript. We appreciate your constructive comments. We have made the following revisions in response to the issues you raised.

1. Since it is not possible to evaluate an independent validation cohort, and given the limited number of tested samples and consequently the poor number of clinical events analyzed, findings from this report should be interpreted with more caution, and thus a mention of this important limitation in the discussion is not enough. Authors should avoid to extend their finding to all TNBC patients, despite, they limited their results to their "cohort" in some sections but not through the entire manuscript as seen by the tittle, and conclusion sections, this need to be addressed.

Response: As you mentioned, our study included a relatively small number of patients with TNBC. Therefore, the title of the present study includes “Results from a retrospective single-center study”. In addition, as you suggested, we added “of the present cohort” in the Conclusion section. 

“This study clearly suggests that myxoid change and FF are independent poor prognostic indicators in patients with TNBC of the present cohort.” (page 27, line 377)

2. An interesting data is presented by the comparison of myxoid change between pre-operative biopsy specimens and surgically-resected specimens. Can you please add a brief commentary on how this changes are established. How therapeutical interventions can cause this phenomena?

Response: As you suggested, we added comments on the mechanism of myxoid change and the effect of chemotherapy. 

“Furthermore, the detailed mechanism of myxoid change in the tumor stroma, which represents hyaluronan accumulation, is unclear, although the patterns of stromal reaction are recognized to be associated with tumor microenvironment, including immune cells and cancer-associated fibroblasts [6]. Neoadjuvant chemotherapy may affect the tumor microenvironment, which may lead to a change of the stromal reaction of TNBC.” (page 26, line 359–page27, line 364)

3. Since the "Presence of myxoid change and FF was significantly associated with low/intermediate TILs in the stroma (p=0.013)", what is the advantage to test FF and myxoid change over TILS, which is now a days an standardized pathologic test made on clinical samples, and more robust international guidelines had being described for its evaluation and interpretation. An although some controversial results, this immune measurement has been also related with prognosis and survival outcomes. Please comment this.

Response: As you mentioned, our study results showed that presence of myxoid change and FF was significantly associated with low/intermediate TILs in the stroma. Evaluation of TILs in breast cancer specimens is performed in clinical practice. Although our study suggests that analysis of myxoid change and FF is a practical and easily assessable method, further studies are needed to verify the significance of myxoid change and FF, compared to analyzing TILs.

---

## [Decision Letter · Decision Letter 2]

7 Jan 2021

Presence of myxoid stromal change and fibrotic focus in pathological examination are prognostic factors of triple-negative breast cancer: Results from a retrospective single-center study.

PONE-D-20-23411R2

Dear Dr. Ishida,

We’re pleased to inform you that your manuscript has been judged scientifically suitable for publication and will be formally accepted for publication once it meets all outstanding technical requirements.

Kind regards,

Elda Tagliabue

Academic Editor

PLOS ONE

Additional Editor Comments (optional):

Reviewers' comments:

Reviewer's Responses to Questions

**Comments to the Author**

1. If the authors have adequately addressed your comments raised in a previous round of review and you feel that this manuscript is now acceptable for publication, you may indicate that here to bypass the “Comments to the Author” section, enter your conflict of interest statement in the “Confidential to Editor” section, and submit your "Accept" recommendation.

Reviewer #1: All comments have been addressed

Reviewer #2: All comments have been addressed

2. Is the manuscript technically sound, and do the data support the conclusions?

Reviewer #1: Yes

Reviewer #2: Yes

3. Has the statistical analysis been performed appropriately and rigorously? 

Reviewer #1: Yes

Reviewer #2: Yes

4. Have the authors made all data underlying the findings in their manuscript fully available?

Reviewer #1: Yes

Reviewer #2: Yes

5. Is the manuscript presented in an intelligible fashion and written in standard English?

Reviewer #1: Yes

Reviewer #2: Yes

6. Review Comments to the Author

Reviewer #1: The authors have addressed the queries raised in my previous review.

The manuscript in the present form is suitable for publication.

Reviewer #2: The authors addressed most of the points, so I consider the text is now suitable to be published in the journal

7. PLOS authors have the option to publish the peer review history of their article (what does this mean?). If published, this will include your full peer review and any attached files.

Reviewer #1: No

Reviewer #2: No

---

## [Editor Report · Acceptance letter]

13 Jan 2021

PONE-D-20-23411R2 

Presence of myxoid stromal change and fibrotic focus in pathological examination are prognostic factors of triple-negative breast cancer: Results from a retrospective single-center study 

Dear Dr. Ishida:

I'm pleased to inform you that your manuscript has been deemed suitable for publication in PLOS ONE. Congratulations! Your manuscript is now with our production department. 

Kind regards, 

on behalf of

Dr. Elda Tagliabue 

Academic Editor

PLOS ONE